# Simulation and Analysis of the Dynamic Characteristics of Groundwater in Taliks in the Eruu Area, Central Yakutia

Miao Yu [1,2,3], Nadezhda Pavlova [4,*], Changlei Dai [2,3], Xianfeng Guo [5], Xiaohong Zhang [2,3,6], Shuai Gao [2,3,7] and Yiru Wei [1,2,3]

1   Faculty of Geology and Survey, M. K. Ammosov North-Eastern Federal University, Yakutsk 677000, Russia; hss_yumiao@126.com (M.Y.)
2   School of Hydraulic & Electric-Power, Heilongjiang University, Harbin 150080, China
3   International Joint Laboratory of Hydrology and Hydraulic Engineering in Cold Regions of Heilongjiang Province, Harbin 150080, China
4   Melnikov Permafrost Institute of the Siberian Branch of the Russian Academy of Science, Yakutsk 677000, Russia
5   Department of Intelligent Agriculture, Heilongjiang Agriculture Economics Vocational College, Mudanjiang 157041, China
6   Institute of Natural Sciences, M. K. Ammosov North-Eastern Federal University, Yakutsk 677000, Russia
7   Institute of Engineering & Technology, M. K. Ammosov North-Eastern Federal University, Yakutsk 677000, Russia
*   Correspondence: napavlova@mpi.ysn.ru

**Abstract:** The perennially unfrozen zones (taliks) in the Eruu area of central Yakutia have a complex stratigraphic structure, and the dynamic characteristics of groundwater in this region have been insufficiently studied. This study analyzed the results of the explorations and geophysical studies conducted by the Melnikov Permafrost Institute of the Siberian Branch of the Russian Academy of Science. In addition, we simulated and analyzed the dynamic characteristics of groundwater in the area based on hydro-meteorological data, snow data, and remote sensing data. During the process, the dynamic changes in the attributes of aquifers due to the seasonal freeze–thaw processes of soils, including the active layer, were also taken into account. The results showed the following: (1) According to the analysis of the measured data on water levels in hydrogeological observation well 14E/2014, the difference between the simulated and measured values of groundwater levels in monitoring wells for over 99% of the measurements was less than 0.1 m. The average difference between the measured (excluding missing values) and simulated values of groundwater level in the monitoring wells was 0.028 m/d. (2) The annual average water level in the study area declined. The simulated value dropped at a rate of 0.10 m/a, with only a gap of 0.01 m/a with the measured value. Meanwhile, the simulated water head was greatly influenced by the terrain, especially in the central area, where the head decreased rapidly from the perimeter toward the lakes (8.9 m/km on average). (3) From 1 September 2014 to 31 August 2015, the mean value of the simulated discharge in the study area was 3888.39 L/d, which was in line with the results of previous monitoring (the average flow was 4147.20 L/d and 3715.20 L/d in 2014 and 2015, respectively). This study can provide a reference for the reasonable exploitation and utilization of groundwater under the influence of the distribution of perennially unfrozen zones, or taliks, and provides an effective three-dimensional modeling method for quantifying the analysis of groundwater dynamics in permafrost regions.

**Keywords:** groundwater level; dynamic characteristics; talik; permafrost; MODFLOW-USG

## 1. Introduction

Permafrost (defined as the ground where the temperature remains below 0 °C for at least two consecutive years) is a key component of the cryosphere [1,2]. As permafrost is a relatively impermeable layer, at certain spatial and temporal scales it impedes the

hydraulic connection between surface water and groundwater [3]. In addition, the seasonal freeze–thaw cycles of the active layer significantly affect the direction, velocity, and circulation pattern of groundwater seepage, which results in the fact that the groundwater transport theories and mechanisms used in some unfrozen zones are not applicable in frozen zones [4–6]. In permafrost regions, the permafrost constitutes a relatively stable regional impermeable layer, which changes the structure of groundwater from a single layer to one that is two or even three layers [7]. Sub-permafrost aquifers are constrained by the seasonally thawed layer. They are thin, shallowly buried, only recharged by infiltration of atmospheric precipitation, and at the same time influenced by evaporation, resulting in unstable water levels and phases, seasonal changes in water quantity, and water quality that is susceptible to pollution [8]. As a result, it is difficult to use them as long-term and stable freshwater resources [9,10]. Infrapermafrost water, which is restricted by the upper permafrost, has very poor recharge sources, with slow runoff and obstructed discharge [11]. Meanwhile, as groundwater stays in the aquifer for a long time, its water quality is complex, the hydrochemical type is variable, and mineralization is generally high [12,13]. Its distribution and occurrence patterns are extremely complex, as they are controlled by many factors, such as permafrost distribution, talik distribution, and the distribution of fault structures [11]. As a result, the quantity and quality of water can vary widely, and it is generally difficult to find water-rich zones with satisfactory water quantity and quality [14].

In central Yakutia, the demand for freshwater is gradually increasing as population growth leads to greater consumption of clean drinking water; thus, more interpermafrost water, including that in the Eruu spring area, is exploited [15]. However, the special nature of drilling operations under inclement climatic conditions in permafrost zones, and the periodic freezing of water in the wells, create many technical difficulties related to the research on groundwater in frozen zones. The hydrogeological numerical model is highly adaptable and can accommodate varying geological attributes, geometries, and boundary conditions [16]. Therefore, the use of numerical simulations to assess the characteristics of aquifers in permafrost zones has received increasing attention [17–19].

Generally, creating a hydrogeological numerical model requires defining the temporal and spatial properties of the model, including setting the starting time and the vertical, horizontal, and lateral boundaries [6,20]. The continuous space is divided into finite blocks, and the continuous time is divided into finite time steps [21]. The physical and mechanical properties of the geological formations are then specified, followed by designating the groundwater seepage conditions as a function of time [22]. After running the model, a new water level distribution is calculated for each time step based on the combination of water level, prerequisite, and boundary conditions [19,23]. In constructing numerical models for multi-year permafrost regions, the seasonal changes in active layer geological stratum attributes [24], the insulating effects of snow cover on the soil surface [25], and soil moisture changes resulting from snow-melt infiltration make the modeling process more complex [26,27]. Frampton employed a two-dimensional water–heat coupling model to study the processes of groundwater seepage and discharge in multi-year permafrost regions. The modeling results suggest that as the thickness of the island-shaped active layer in these regions increases, more groundwater infiltrates into deeper aquifers and is later expelled onto the surface, thereby amplifying the depth and length of the groundwater flow paths [28]. Breemer used the MODFLOW groundwater model to simulate the two-dimensional groundwater seepage processes in the Lake Michigan Lobe region and achieved simulation results for steady-state hydraulic head values and surface drainage volumes that were comparable to those measured in the field [29]. Wellman utilized simulations to analyze the two-dimensional groundwater seepage processes in the talik zone of Alaska's interior Yukon region and demonstrated how time, climate, lake size, and hydraulic conditions influenced the development of taliks and variations in water content within the aquifer [30]. Nevertheless, most existing models do not explicitly consider the seasonal freeze–thaw processes in soils, including the active layer or the impact of freeze–thaw processes on hydrogeological conditions. In addition, some models only address one-

or two-dimensional problems with relatively coarse spatial resolution [22,28]. Therefore, the existing studies still present great uncertainties.

In this study, MODFLOW-USG (unstructured grid), a three-dimensional standard groundwater seepage model, was used to quantitatively assess the dynamic characteristics of groundwater in the Eruu area, a typical talik in central Yakutia, with the dynamic changes in the attributes of aquifers in different seasons taken into account. This study can provide a reference for the reasonable exploitation and utilization of groundwater under the influence of the distribution of perennially unfrozen zones, or taliks, and provides an effective three-dimensional modeling method for quantifying the analysis of groundwater dynamics in permafrost regions.

## 2. Study Area and Data Pre-Processing

Firstly, data on the surface elevation, precipitation, snow depth, permafrost distribution, talik distribution, and spring flow were collected before the model was built [18,31]. Secondly, a MODFLOW-USG model considering the impact of the distribution of permafrost and taliks was constructed. Thirdly, the model was applied to the study area. Lastly, the dynamic characteristics of groundwater were analyzed.

### 2.1. Overview of the Study Area

Bestyakh is located on the east bank of the Tama River in Megino-Kangalassky, the central part of the Sakha (Yakutia) Republic, Russia, 12 km from the estuary. In 1976, the Siberian Branch of the Russian Academy of Sciences, based on surveys, observations, drilling, laboratory filtration, and hydrochemistry, determined the catchment area in the region as being 9.21 km$^2$. This is a zone of continuous permafrost with a thickness of 300 m or more. Sand ridges from 3 to 20 m high are widely distributed on the terrace surface. In addition, this rugged area has many stream valleys, small rivers, and lake basins.

The study area located at the foot of sand deposits in the terrace is an area of year-round icy area of Eryu formed by interpermafrost water, where the terrain slopes from southwest to northeast. The absolute elevation of the surface gradually decreases from 163 to 140 m, and the interpermafrost talik is the main water supply source. The tubular talik ranges from 500 to 900 m wide, the top and bottom of which are at depths of 13–53 and over 55.4 m from the Earth's surface, respectively. There are three small lakes, Bosogor, Ergen, and Abaga-Quill the talik (Figure 1). In 2007, the hydrogeological observation well 1-2007 was drilled within the groundwater recharge zone between Ergen Lake and Abaga-Quill Lake [32]. Interlayer water in the frozen layer was found to be pressurized at depths of 23.9–32.0 m, and water samples collected from the lakes, talik aquifer, and Eruu spring were found to be chemically identical. In 2014, the hydrogeological observation well 14E was drilled between Abaga-Quill Lake and Eruu spring. Beneath the permanent permafrost at a depth of 13 m, water-saturated sand at 0.2 °C was discovered. At the point where the groundwater was discharged, the thickness of the aquifer was determined to be 23–30 m. The aquifer consisted primarily of fine and medium-grained sand of the Quaternary period, while the aquifer at the springwater edge was composed of Quaternary sandstone, gravel and pebble deposits, and upper Jurassic sandstone [15]. The study area is covered by permafrost, and the active layer is 2–4 m thick [15]. When the active layer is not completely frozen, the recharge sources are mainly lakes and atmospheric precipitation. In general, snow melting starts to recharge the groundwater in April each year, while interpermafrost water plays this role throughout the year on the southwest side. On the northeast side, part of the groundwater is discharged as interpermafrost water, and another part is discharged out of the surface in the form of springs, resulting in ice accumulation.

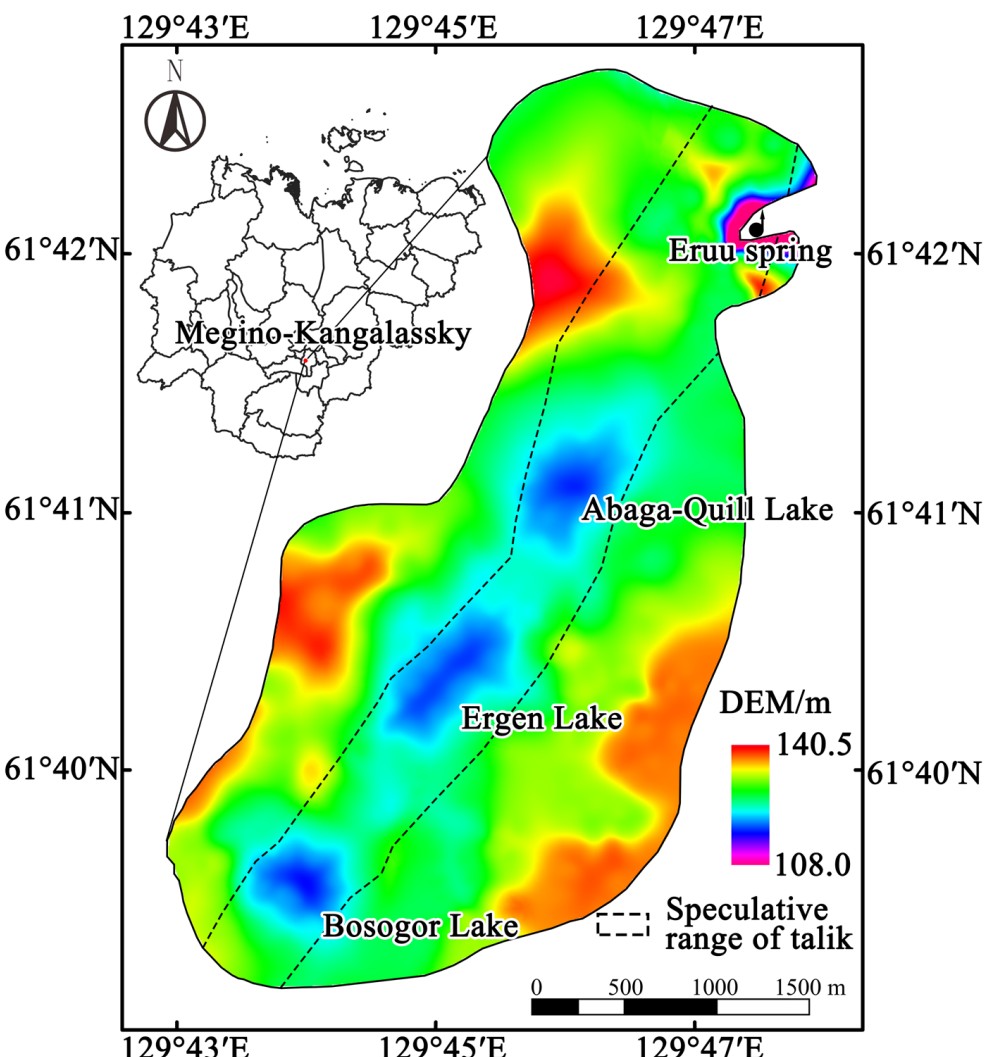

**Figure 1.** Surface elevation and spatial location of the Eruu spring area (The DEM data used in this study were obtained from the ASTER GDEM V3 dataset published by NASA in 2019. The data have a spatial resolution of approximately 30 m and were calibrated using actual topographic data measured by the Melnikov Permafrost Institute of the Siberian Branch of the Russian Academy of Science).

*2.2. Data Source and Pre-Processing*

The digital elevation model (DEM) data were obtained from the National Cryosphere Desert Data Center. The solar radiation data were obtained from the Pokrovsk hydrometeorological station, which is the closest to the study area. In line with the latitude, longitude, and altitude conditions, the temperature and precipitation data obtained from three representative hydrometeorological stations near the study area (Pokrovsk, Yakutsk, and Tegyulta) were selected, and these data were from the All-Russian Research Institute of Hydrometeorological Information. Data from the three stations cannot directly reflect the temperature and precipitation conditions in the study area because of the relatively long distances to the study area, despite the small differences in terms of altitude. Therefore, before the analysis, the data on the study area needed to be corrected based on the temperature data from the stations. The inverse weighting distance (IDW) method was used to correct the data for distance, and the interpolation points and the spacing of sample points were used as weights for the weighted average. The equation is as follows [33]:

$$X = \left( \sum_{i=1}^{n} \frac{X_i}{D_i^p} \right) / \left( \sum_{i=1}^{n} \frac{1}{D_i^p} \right) \tag{1}$$

where *X* is the temperature or precipitation of the study area (°C or mm); $D_i$ is the horizontal distance between the interpolated hydrometeorological station and the study area (m); $X_i$ is the temperature or precipitation of the interpolated hydrometeorological station (°C or mm); *n* is the number of interpolated sample points; and *p* is the power exponent used to calculate the weight of distance.

## 3. Construction of Simulation

### 3.1. Construction of the Conceptual Hydrogeological Model

MODFLOW is a standard three-dimensional finite-difference groundwater model. Many studies have used MODFLOW in temperate regions, and only a few have used it in cold regions [34]. The size of the study area was 9.21 km$^2$. The simulated calculation area was treated as active cells, and the area out of the boundary of the calculation area was treated as inactive cells. In any calculation related to flow, water level, etc., the inactive grid cells are ignored by the model and are not involved in the calculation. When the grid was dissected, the study area was included and divided into three-dimensional grids with a horizontal resolution of 15 m × 15 m in the X- and Y-directions. In the Z-direction, each column was divided into six layers. The top layer was the active layer, which consisted mainly of saturated shallow groundwater and the aeration zone. As this layer is affected by seasonal freezing and thawing, the Modflow-usg that supports time-varying aquifer properties was used.

The following conceptualization of the lateral boundaries was used: Interpermafrost water in the southwestern part of the study area was the source of water, which was conceptualized as the inflow boundary. The terrain of the northeastern region is relatively low, where part of the groundwater discharges as interpermafrost water, and another part discharges out of the surface in the form of springs, which were conceptualized as the outflow boundary. Permafrost is mainly distributed in the northwest and southeast regions of the study area, which was conceptualized as the confining boundary (Figure 2). The following conceptualization of the vertical boundaries was used: as the active layer at the top had vertical water exchange with the outside in different ways, such as rainfall infiltration, snowmelt infiltration, evapotranspiration, and exchange with lake water, the top of the study area had the free water surface of the active layer as the upper boundary of the system. Due to the seasonal freezing and thawing of the active layer, its thickness and the properties of the aquifer changed continuously throughout the year [35–37]. Therefore, we used the Time-Variant Materials package (TVM) to simulate the properties of the active layer, which enabled the model to change these properties over time while maintaining spatial discretization. The impact of evapotranspiration on vertical water exchange was ignored in this simulation because it is small. At the bottom of the system, the impermeable layer composed of permafrost, etc., served as the bottom boundary.

We conducted electrical resistivity tomography scanning to survey the profiles of two taliks in the study area, and the results are presented in Figure 3. Within the surveyed sections, the measured electrical resistivity ranged from 50 to 50,000 Ω·m [15]. Combined with previous experimental data, the unfrozen soil layers had an electrical resistivity range of 50–2000 Ω·m. The electrical resistivity range of frozen soil layers at a temperature near −0.2 °C was 2000–10,000 Ω·m, while the electrical resistivity range of permafrost at lower temperatures was 10,000–50,000 Ω·m. The water-bearing layer detected in profile $A_1$–$A_2$ had an estimated width of about 650 m, with a top depth of 13–30 m. The water-bearing layer detected in profile $B_1$–$B_2$ had an estimated width of about 500 m, with a top depth of 20–34 m.

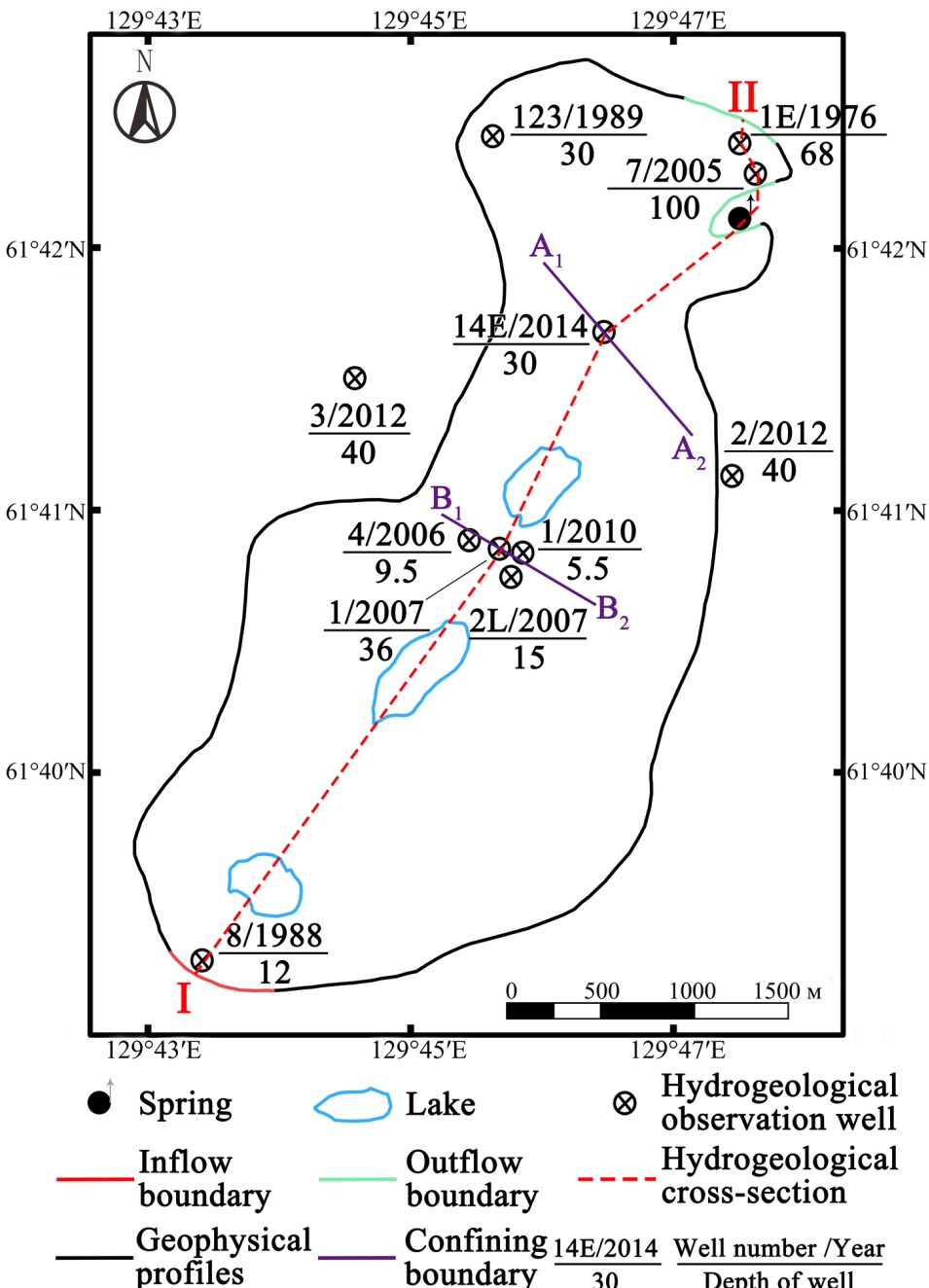

**Figure 2.** Model boundary conditions and distribution of hydrogeological observation wells in the study area.

The study area was surveyed, observed, drilled, and explored, and a geophysical study was carried out with electromagnetic tomography. The geological profile of the study area is shown in Figure 4. The model was stratified according to the research results. Layer 1 is the active layer; Layers 2 and 6 mainly consist of permafrost, with the temperature of the permafrost layer typically ranging from −0.7 to −0.02 °C; Layer 3 mainly consists of quartz–feldspar fine- and medium-grained sand; Layer 4 mainly consists of alluvial gravel–pebble deposits of igneous and sedimentary rocks; Layer 5 mainly consists of fine- and coarse-grained sandstone with siltstone interlayers. The stratigraphic structure of a typical hydrogeological observation well in the study area is shown in Figure 5.

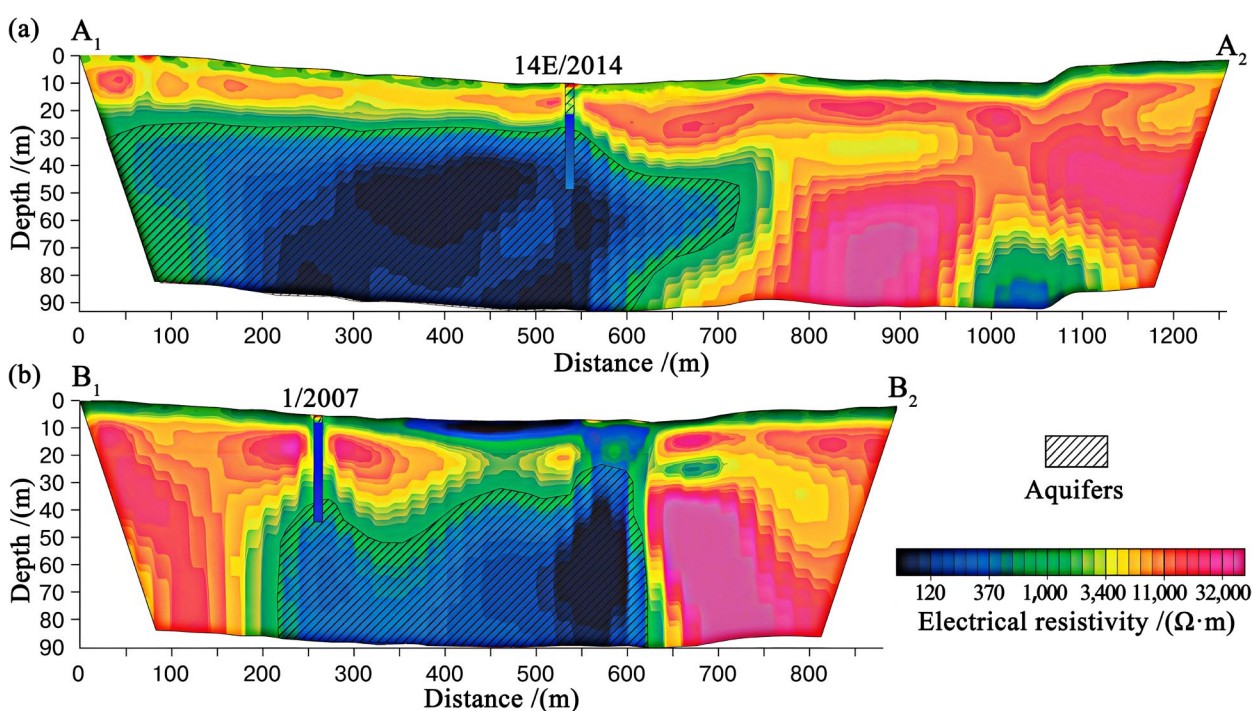

**Figure 3.** Geophysical section of the study area ((**a**) lines A1–A2, (**b**) lines B1–B2 in Figure 2) [15].

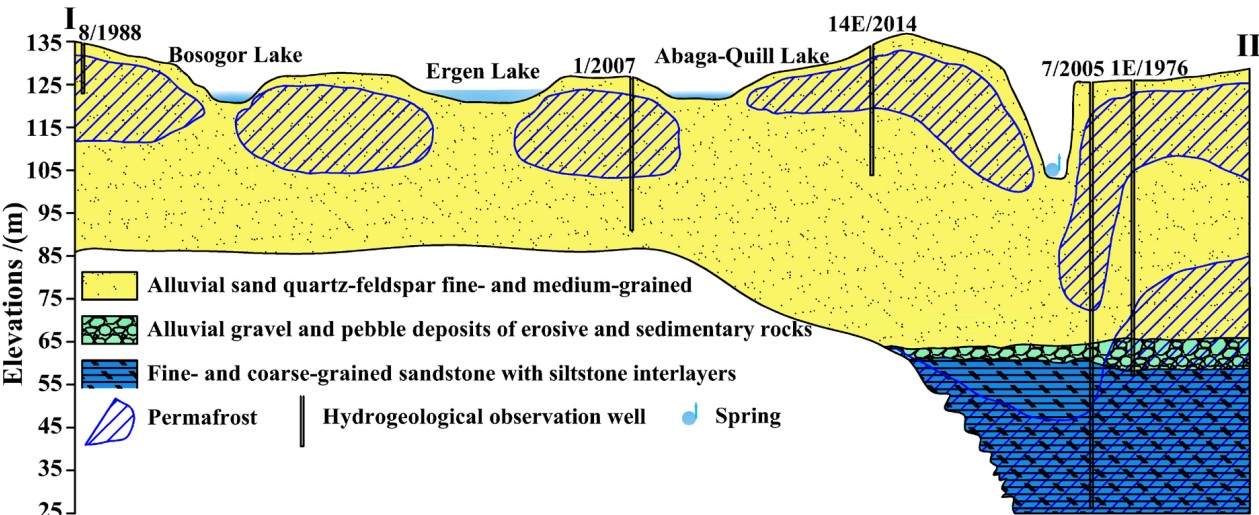

**Figure 4.** Hydrogeological sections of the study area (lines I–II in Figure 2).

For the elevation of each stratum, 1208 points were extracted from the elevation value for interpolation. The surface elevation was extracted mainly from the data of the digital elevation model (DEM); the rest of the elevation values of other layers were obtained from the borehole data provided by the Melnikov Permafrost Institute of the Siberian Branch of the Russian Academy of Science; and the data on the exploration of the strata were obtained using electromagnetic tomography. The modeling process is shown in Figure 6.

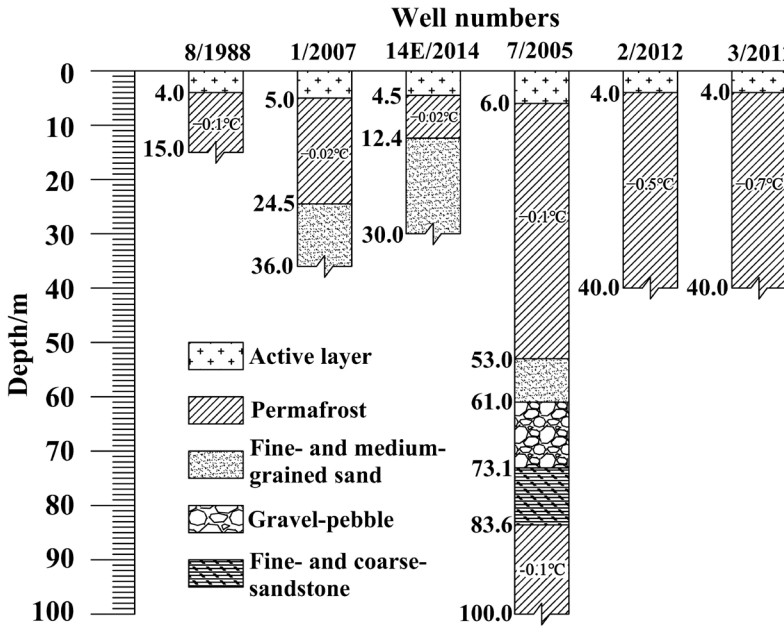

**Figure 5.** The stratigraphic structure of a typical hydrogeological observation well in the study area.

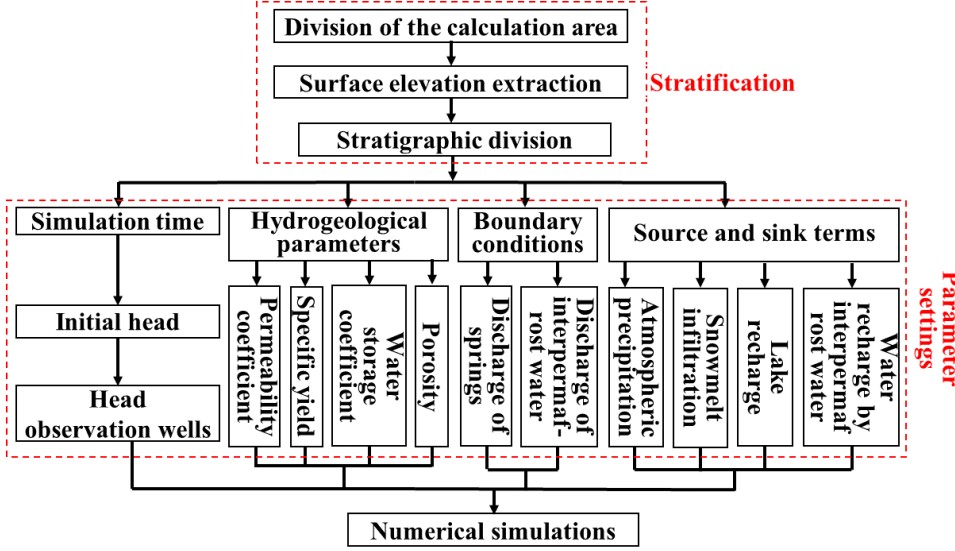

**Figure 6.** Modeling process.

*3.2. Construction of the Groundwater Mathematical Model*

3.2.1. Mathematical Model

According to the laws of groundwater movement, such as the principle of mass conservation and Darcy's law, as well as the scientific and systematic analysis of relevant parameters such as hydrogeological conditions and permeability coefficient and groundwater flow field in the study area, the following set of partial differential equations and boundary conditions were used to express the conceptualized groundwater flow system in the study area [38]:

$$\frac{\partial}{\partial x}\left(K_x\frac{\partial h}{\partial x}\right) + \frac{\partial}{\partial y}\left(K_y\frac{\partial h}{\partial y}\right) + \frac{\partial}{\partial z}\left(K_z\frac{\partial h}{\partial z}\right) + w = \mu_s\frac{\partial h}{\partial t}, \quad (x, y, z) \in \Omega, \quad h = z, \quad (2)$$

$$\frac{\partial}{\partial x}\left(K_x\frac{\partial h}{\partial x}\right) + \frac{\partial}{\partial y}\left(K_y\frac{\partial h}{\partial y}\right) + \frac{\partial}{\partial z}\left(K_z\frac{\partial h}{\partial z}\right) = S_s\frac{\partial h}{\partial t}, \quad (x, y, z) \in \Omega \qquad (3)$$

$$H(x, y, z, 0) = H_0(x, y, z), \in \Omega \tag{4}$$

$$q(x, y, z)|\Gamma_3 = k'\frac{h - h_0}{B'}, \ h \geq h_0, \ (x, y, z) \in \Gamma_3 \tag{5}$$

where $\Omega$ is the simulation area; $h$ is the groundwater level (m); $w$ is the vertical water exchange (m$^3$/(d · m$^2$)); $H_0$ is the initial water elevation (m); $x$, $y$, and $z$ are coordinate variables (m); $Kx$, $Ky$, and $Kz$ are the permeability coefficients in the $x$, $y$, and $z$ directions (m/d); $\mu_s$ is the specific yield; $S_s$ is the elastic storativity (1/m); $t$ is the time (d); $q$ is the third-type boundary discharge per unit width (m$^3$/d); $k'$ is the boundary permeability coefficient (m/d); $h_0$ is the boundary control head (m); $B'$ is the boundary length (m); $\Gamma_3$ is the discharge boundary of the study area.

Equation (2) is the continuity equation of groundwater movement (i.e., the equation of hydraulic balance);

Equation (3) is the Boussinesq equation, which represents three-dimensional seepage;

Equation (4) is the water elevation at the moment when the initial water elevation is 0;

Equation (5) is a third-type boundary condition (i.e., spring discharge).

### 3.2.2. Boundary Conditions

According to the conceptual model, the bottom, the northwest side, and the southeast side of the model are confining boundaries. The top layer of the study area is recharged by rainfall, lake water, and snowmelt infiltration. Data on rainfall and snowmelt infiltration were input by the Recharge module, and lake infiltration data were input by the LAK module. The southwest side, recharged by interpermafrost water, was input by the RCH module. The discharge boundary consisted of two parts. One part is discharged to the interpermafrost layers on the northeast side, which was processed by the Drain module; the other part is mainly discharged in the form of springs. The elevation of a spring mouth was fixed. When the groundwater level was higher than the elevation of the spring mouth, the groundwater was discharged to the outside through the spring. When the groundwater level was lower than the elevation of the spring mouth, the spring no longer discharged groundwater to the outside because of the cutoff. Therefore, the exchange volume was defined as follows:

$$Q_{spr} = C_{spr}(h - h_{spr}) \tag{6}$$

where $Q_{spr}$ is the exchange volume between the spring and groundwater (m$^3$/d); $C_{spr}$ is the flow conduction coefficient between the spring and groundwater aquifers (m$^2$/d); $h$ is the groundwater level (m); and $h_{spr}$ is the elevation of the spring mouth (m).

### 3.2.3. Hydrogeological Parameters

The initial hydrogeological parameters of aquifers were obtained from a multi-year study by the Melnikov Permafrost Institute of the Siberian Branch of the Russian Academy of Science. When the active layer is completely frozen in winter, its coefficient of transmissivity is extremely low, close to that of permafrost. As the hydraulic properties of the active layer vary at different depths and are intricately interrelated with thermal properties, "effective" aquifer properties are considered to be highly nonlinear [39]. Therefore, we used nonlinear interpolation to estimate the time series of "effective" aquifer properties.

### 3.2.4. Source and Sink Terms

Groundwater recharge and discharge are the basic factors that determine groundwater cycles and affect the formation of groundwater runoff. Different sources of recharge and discharge, coupled with spatial and temporal variations in recharge and discharge, directly affect the process of groundwater runoff and the dynamic changes in water quantity. There are three main types of recharge sources in the study area, namely groundwater recharge on the southwest side, atmospheric precipitation, and lake recharge and snowmelt infiltration

recharge. On the southwest side, groundwater in taliks recharges the study area, and the water level data of well No. 8/1988 were used to measure the amount of this type of recharge. The amount of atmospheric precipitation recharge in summer was represented by an effective rainfall infiltration coefficient $\alpha$, and $\alpha = 0.85$. The LAK module was used to address the water exchange between the lake and groundwater. For snowmelt infiltration in spring, an equation [40] was used to calculate the amount of snowmelt infiltration per day, and the calculated values were input into the following model:

$$M = m_Q R_d + a_r T_d, \tag{7}$$

where $M$ represents the amount of snowmelt infiltration (cm·d$^{-1}$), $m_Q$ is the physical constant of energy conversion into the depth of snowmelt water (here 0.026 cm·W$^{-1}$·m$^2$·d$^{-1}$), $R_d$ is the net radiation index (W·m$^{-2}$), and $a_r$ is the modified degree–day factor (here 0.23 cm·°C$^{-1}$·d$^{-1}$). $T_d$ (°C) is the degree–day index.

## 4. Results and Analysis

The impact of permafrost distribution on groundwater dynamics was estimated using the parametric MODFLOW-USG model. The simulation lasted from 1 September 2014 to 31 August 2019, with a time step of 1 d. In this case, the initial water level was based on the borehole data of 2014.

### 4.1. Model Identification and Analysis

Model identification means continuously adjusting the relevant parameters in the model, and after each adjustment, using the model to simulate the groundwater level [41]. The difference between the simulated level and the measured water level was analyzed. Another adjustment was required when the difference was large, and this process stopped when the difference met the requirements (i.e., the simulated curve and the measured water level best fit each other) [42].

Due to the remote location of the study area and the continuous harsh negative temperature conditions, the 10 hydrogeological observation wells missed some data measurements during the long-term monitoring of groundwater levels. The measured data of groundwater levels in well 14E/2014 from 1 September 2014 to 31 August 2019, which were relatively complete and accurate, were selected as the data for numerical model identification (Figure 7). Parameters such as permeability coefficient and specific yield in the model were continuously adjusted and calibrated to make the simulation results of the groundwater levels fit well with the measured values, and the dynamic changing trends of the two were largely consistent with each other.

During the period of model identification, the analysis and comparison of the simulation results of all monitoring wells in the study area and the available measured data showed that the errors of fitting for all of the wells were less than 0.1 m. It was found that the error between the measured and simulated data was large from late April to late May every year because, during that period, the active layer melts. The temperature during the day has positive values, and at night has negative values. The surface of the active layer experiences frequent alternations between freezing and melting, resulting in large fluctuations in the groundwater level. The maximum error between the simulated and measured values occurred on 29 April 2015, reaching 0.26 m. The difference between the simulated and measured values of the groundwater level in monitoring wells for over 99% of the measurements was less than 0.1 m. The average difference between the measured (excluding missing values) and simulated values of the groundwater level in monitoring wells was 0.028 m/d, which truly reflected the hydrogeological parameters of the conceptual hydrogeological model in the study area. The values of the optimized parameters are shown in Table 1.

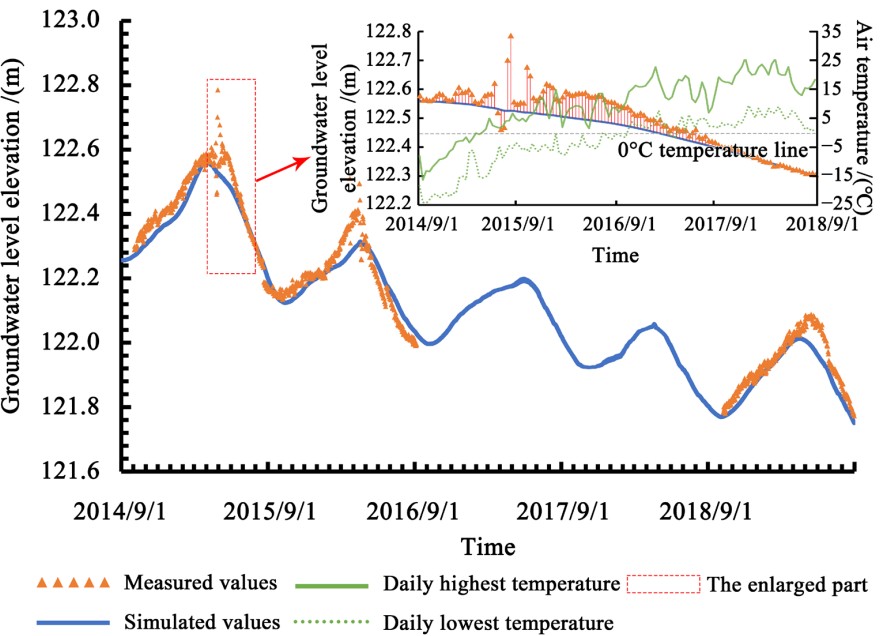

**Figure 7.** Dynamic changes in the water level seen in well 14E/2014 in the study area, 1 September 2014 to 31 August 2019 (with the part from 1 April 2015 to 1 August 2015 enlarged).

**Table 1.** The ranges in initial parameter values for each layer.

| Layer | Horizontal Permeability Coefficient | Vertical Permeability Coefficient | Specific Yield |
|---|---|---|---|
| Active layer | $8.0 \times 10^{-5}$~6.5 | $1.2 \times 10^{-5}$~2.67 | $0.21 \times 10^{-5}$~0.18 |
| Permafrost | $8.0 \times 10^{-5}$ | $1.2 \times 10^{-5}$ | $0.21 \times 10^{-5}$ |
| Fine- and medium-grained sand | 4.25~6.50 | 1.25~2.25 | 0.12~0.14 |
| Gravel–pebble | 113.08 | 40.50 | 0.20 |
| Fine- and coarse-sandstone | 14.55 | 4.85 | 0.06 |

*4.2. Dynamic Changes in the Groundwater Level*

Taliks after the Holocene are preserved in the study area, which are 20 to 50 m deep and located below the permafrost of the Pleistocene. As shown in Figure 7, there are three lakes above the taliks; the frequent water cycles among the lakes regulate the dynamic changes in groundwater to some extent, which are seasonal and perennial in nature. The reason is that the active layer under the lakebed gradually freezes over time when the water levels of the lakes fall in winter as the water quantity reduces. The active layer in some way hinders the infiltration and recharge of lake water to groundwater, and the gaining of water resources in lakes gradually exceeds the loss. It is also the seasonal freezing of the active layer that makes taliks have the characteristics of interpermafrost rock layers. In general, the water level of a lake is higher than the groundwater level. In the following summer, when the surface water level drops, groundwater flows to lakes and replenishes lake water.

During the simulation period, hydrogeological observation well No. 14E/2014 was selected for the dynamic analysis of water levels. The elevation of the wellhead was 134.0 m, the dynamic characteristics of which are shown in Table 2. The annual highest measured water level and the highest simulated water level were 26~30 d apart, and the annual lowest measured water level was no higher than 4 d. The range in the maximum measured annual fluctuation was 0.32~0.82 m, and the range in the maximum simulated annual fluctuation was 0.21~0.36 m. The annual average water level showed a decreasing trend. The measured annual average water level dropped at a rate of 0.11 m/a, and the simulated annual average water level dropped at a rate of 0.10 m/a. The water level in well 14E/2014 was strongly influenced by Abaga-Quill Lake. The decrease in annual precipitation from

1 September 2014 to 31 August 2019 resulted in a reduction in the surface area and water levels of lakes. Consequently, the groundwater level in well 14E/2014 showed a consistent yearly decline. Both the simulated and measured yearly average water levels decreased at a rate of 0.10 m/a. Figure 8 indicates that groundwater levels in the active layer of permafrost areas can respond rapidly to precipitation changes, particularly during the summer. With the exception of a sudden increase in annual precipitation from 1 September 2015 to 31 August 2016, precipitation has steadily decreased each year. In the permafrost areas near the northwest and southeast watersheds, changes in groundwater levels generally followed the trends in precipitation changes.

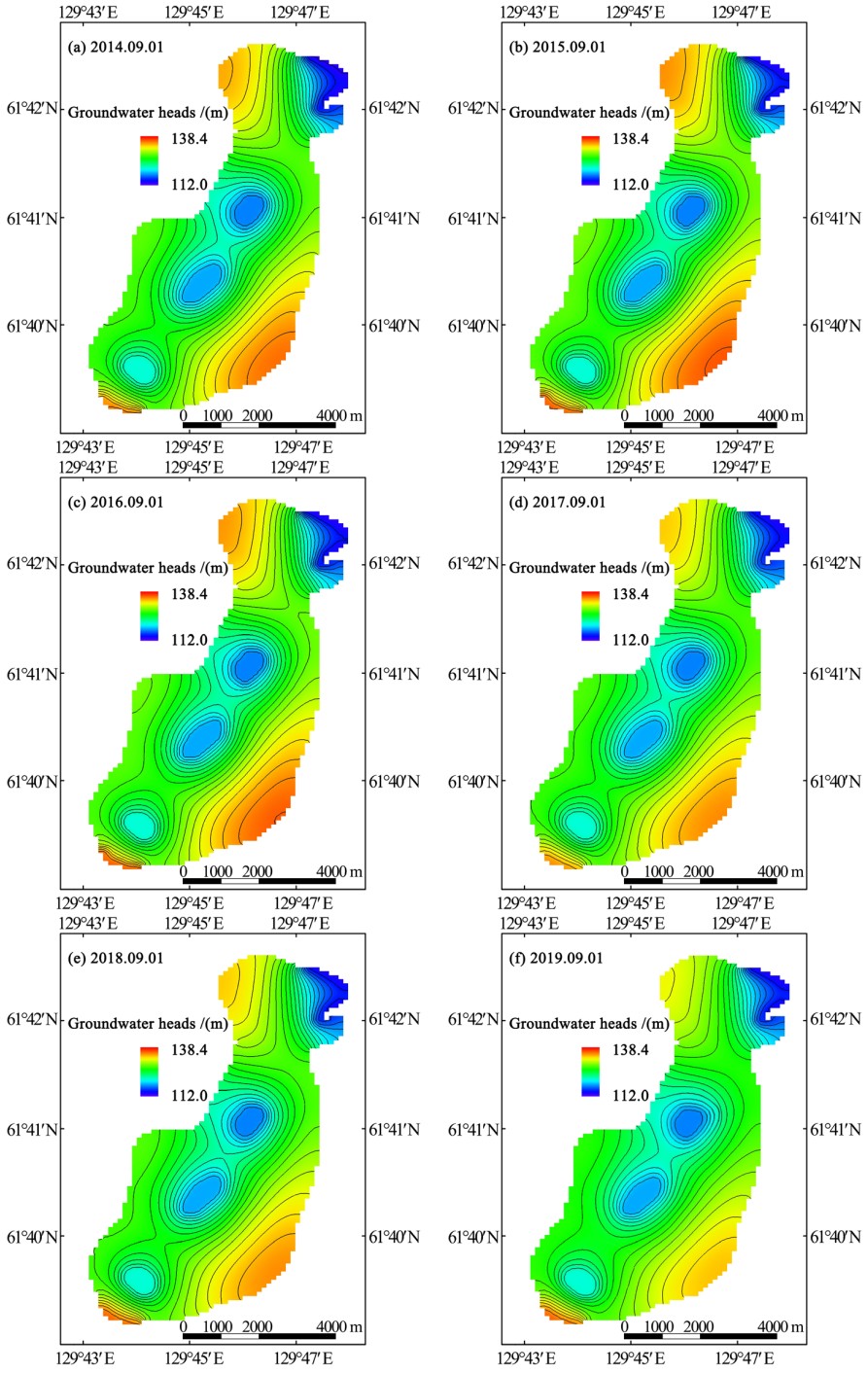

**Figure 8.** Groundwater levels in autumn in the study area, from 2014 to 2019.

**Table 2.** Characteristics of water levels during the freeze–thaw process in hydrogeological observation well No. 14E/2014, from 2014 to 2019.

| Time Interval | Maximum Measured Value/m | Maximum Simulated Value/m | Minimum Measured Value/m | Minimum Simulated Value/m | Measured Annual Average Water Level/m | Simulated Annual Average Water Level/m | Measured Annual Fluctuation/m | Simulated Annual Fluctuation/m |
|---|---|---|---|---|---|---|---|---|
| 1.9.2014–31.8.2015 | 122.79 (2015.4.29) | 122.56 (2015.4.1) | 122.17 (2015.8.31) | 122.20 (2015.8.31) | 122.44 | 122.41 | 0.82 | 0.36 |
| 1.9.2015–31.8.2016 | 122.51 (2016.3.23) | 122.32 (2016.4.18) | 122.00 (2016.8.27) | 122.04 (2016.8.31) | 122.20 | 122.19 | 0.51 | 0.28 |
| 1.9.2016–31.8.2017 | - | 122.23 (2017.5.24) | - | 122.02 (2017.8.31) | - | 122.11 | - | 0.21 |
| 1.9.2017–31.8.2018 | - | 122.08 (2018.4.18) | - | 121.80 (2018.8.31) | - | 121.97 | - | 0.28 |
| 1.9.2018–31.8.2019 | 122.09 (2019.5.16) | 122.01 (2019.4.16) | 121.77 (2019.8.31) | 121.75 (2019.8.31) | 121.90 | 121.89 | 0.32 | 0.26 |

According to the simulation results, groundwater exchange in the study area mainly occurred in Layers 1 to 3, and the simulated water head was strongly influenced by the terrain. In the horizontal direction at the top layer, the water head changed with the surface elevation (Figure 8a). In the southeastern part of the study area, the head changed rapidly under the impact of the terrain. In the central area, the head decreased rapidly from the perimeter toward the lakes (8.9 m/km on average). In the northern area, the head declined gradually from west to east (6.56 m/km on average). In the lake areas, the head was higher than the surface elevation at the lake bottom. The head variation decreased in Layers 2 and 3 due to groundwater flow (Figure 9b,c).

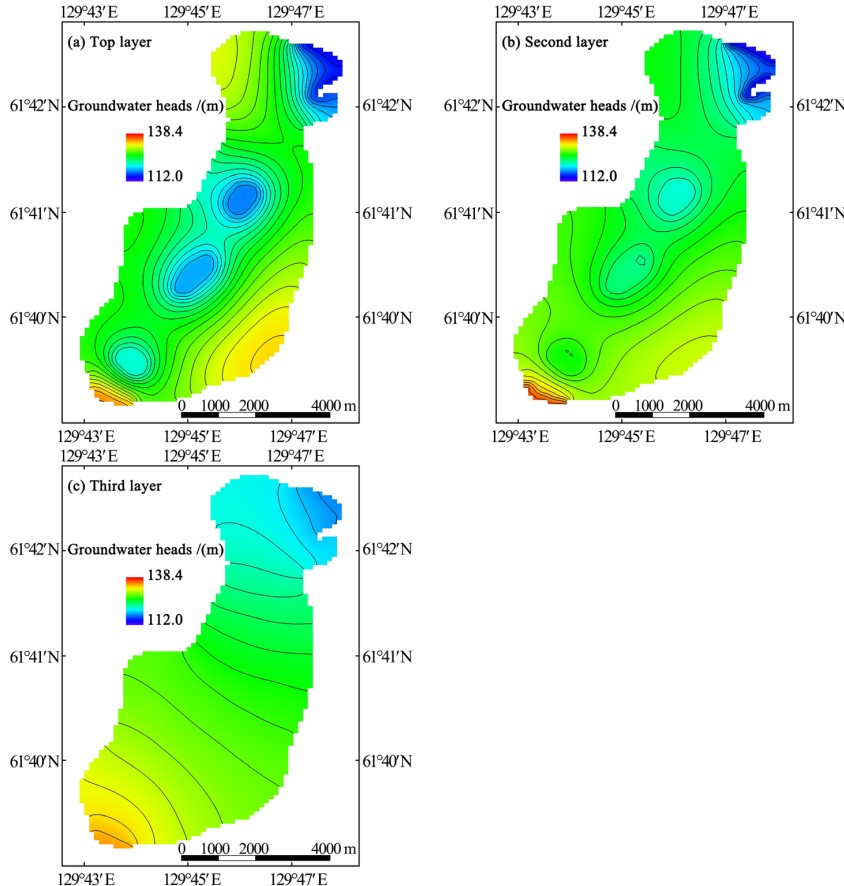

**Figure 9.** 1 September 2019, the spatial distribution of the simulated head in (**a**) the top layer, (**b**) Layer 2, and (**c**) Layer 3, which is similar to the spatial distribution of the surface elevation. The spatial variation of the simulated head in Layer 3 is smooth due to groundwater movement.

Vertically, the head gradient varied with location. The vertical gradient was negative in the northwestern and southeastern areas with higher terrains in the Eruu area and negative near the lakes and in the discharge area. This is because in the summer, groundwater recharges lakes by seepage, and groundwater is discharged to the surface in the form of springs in the discharge area. The northwest and southeast regions received surface water infiltration from atmospheric precipitation.

### 4.3. Dynamic Changes in the Amount of Groundwater Discharge

Based on the field monitoring data and multi-year simulation results, a complete freeze–thaw process of groundwater discharged from the study area from 1 September 2014 to 31 August 2015 was selected (Figure 10). A year was divided into three periods based on the characteristics of seasonal freezing of the active layer: the freezing period (when the temperature is below 0 °C), the thawing period (from when the temperature is below 0 °C to when the ice is completely melted), and the completely unfrozen period. The mean value of simulated discharge during the period was 3888.39 L/d, which was in line with the results of previous monitoring (the average flow was 4147.20 L/d and 3715.20 L/d in 2014 and 2015, respectively) [32].

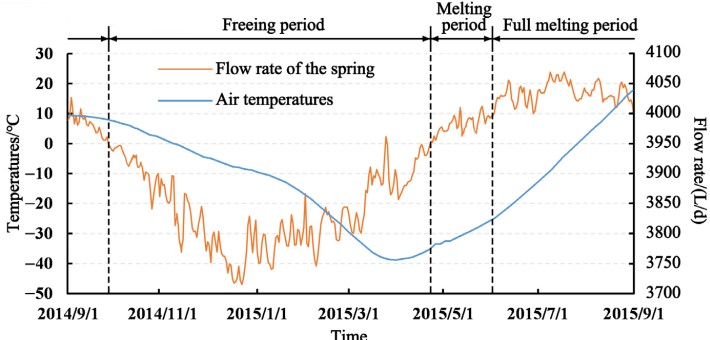

**Figure 10.** Daily average spring flow and the temperature curve from 1 September 2014 to 1 September 2015.

During the freezing period, due to the freezing of the active layer, the discharge area was partially frozen, and the discharge amount decreased. The average discharge during this period was 3889.25 L/d. The lowest value (3755.05 L/d) appeared on 1 April 2015, which led to a rising trend in the groundwater level. Due to the snowfall during this period, the outlet also had a tendency to freeze but was still discharging groundwater under the head pressure. When the groundwater level kept rising and the hydrostatic pressure reached the threshold value, the soil in parts of the discharge area that had not been frozen solid ruptured, and groundwater flowed out with an increasing discharge volume.

During the thawing period, groundwater levels saw significant fluctuations. The temperature during the day had positive values, while that at night had negative values. The discharge area experienced frequent alternations between freezing and melting. At the same time, the snow gradually melted, and because of the flat terrain of the study area, most of the melted snow water saw vertical infiltration. Since the active layer was in the melting process, the snow water gradually began to recharge groundwater, resulting in a slow increase in the discharge volume.

During the completely unfrozen period, the active layer completely melted. As the temperature gradually rose, atmospheric precipitation and water from melting lakes, snow, and the active layer further recharged groundwater. As a result, the discharge rate steadily grew, reaching the maximum value (4037.64 L/d) on 1 September 2015.

## 5. Discussion and Conclusions

### 5.1. Discussion

This study utilized the MODFLOW-USG model to evaluate the dynamic changes in groundwater levels in Eruu, Sakha (Yakutia) Republic, Russia. Considering that the

research area is situated in a region where there is permafrost, the seasonal freeze–thaw processes within the active layer and the infiltration of snowmelt water into the soil during late spring may have an impact on groundwater levels [24,26,27]. To address these impacts comprehensively, the present study undertook a thorough examination of these factors. The issue of dynamic changes in groundwater in permafrost regions has been receiving increasing attention, with some scholars conducting 2D simulation studies to address this problem [28–30]. With the installation of 11 hydrogeological monitoring wells in the study area, we gathered abundant geological data. Building on a profound understanding of boundary conditions, we established a 3D groundwater flow model that is vital to the implementation of sustainable water resource management. Nonetheless, incomplete data caused by the harsh temperature conditions and remote location of the study area were an issue. Thus, we aim to supplement our data further using other water level monitoring methods, including long-term manual observations [43,44].

*5.2. Conclusions*

The simulation results show good applicability of the three-dimensional standard groundwater seepage model MODFLOW-USG that considers the dynamic changes in the attributes of aquifers in different seasons in the Eruu area, as well as relatively good consistency between the simulated groundwater level and spring discharge and the observed data.

(1)　According to the analysis of the measured data on water levels in well 14E/2014, the difference between the simulated and measured values of groundwater level in monitoring wells for over 99% of the measurements was less than 0.1 m. The average difference between the measured (excluding missing values) and simulated values of groundwater level in monitoring wells was 0.028 m/d.

(2)　The annual average water level in the study area declined. The simulated value dropped at a rate of 0.10 m/a, with only a gap of 0.01 m/a with the measured value. Meanwhile, the simulated water head was greatly influenced by the terrain, especially in the central area, where the head decreased rapidly from the perimeter toward the lakes (8.9 m/km on average).

(3)　From 1 September 2014 to 31 August 2015, the mean value of the simulated discharge from the springs in the study area was 3888.39 L/d, which was in line with the results of previous monitoring (the average flow was 4147.20 L/d and 3715.20 L/d in 2014 and 2015, respectively). The discharge process is closely related to various factors, such as the freeze–thaw state, temperature, and hydrostatic pressure of the active layer.

**Author Contributions:** Conceptualization, M.Y. and N.P.; methodology, M.Y.; software, M.Y.; validation, C.D. and X.G.; writing—original draft preparation, M.Y.; writing—review and editing, N.P.; visualization, X.Z., S.G. and Y.W. All authors have read and agreed to the published version of the manuscript.

**Funding:** This research was funded by the Program of Foundation Scientific Research at the Melnikov Permafrost Institute of SB RAS, grant number 122012400106-7 «Groundwater in the cryolithozone: formation and regime patterns, features of interaction with surface water and frozen ground, prospects for use»; China Scholarship Council, grant number 202008230159.

**Institutional Review Board Statement:** Not applicable.

**Informed Consent Statement:** Not applicable.

**Data Availability Statement:** The data presented in this study are openly available and can be downloaded from the following: digital elevation model (DEM) data, http://www.ncdc.ac.cn (accessed on 20 July 2022); temperature and precipitation data, http://aisori-m.meteo.ru/ (accessed on 20 August 2022).

**Conflicts of Interest:** The authors declare no conflict of interest.

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
