# Peer review of "Simulation and Analysis of the Dynamic Characteristics of Groundwater in Taliks in the Eruu Area, Central Yakutia"

_sustainability, doi:10.3390/su15129590_

Round 1
Reviewer 1 Report
Work well presented. Referencing could be improved a little.
Author Response
Response to Reviewer 1 Comments
First of all, I would like to express our sincere gratitude to the reviewers for their comments. Your comments arc all valuable and helpful for revising and improving our manuscript, as well as the important guiding significance to our researches.
Point 1: Work well presented. Referencing could be improved a little.
Response 1: Line 78-105: Within the introduction section, we included more detailed discussion surrounding numerical simulation and positioned our study in relation to previously published literature. Additionally, we bolstered our work with 13 supplementary references. The revised text reads as follows.
The hydrogeological numerical model is highly adaptable, and can accommodate varying geological attributes, geometries, and boundary conditions. Therefore, the use of numerical simulations to assess the characteristics of aquifers in permafrost zones has received increasing attention.
Generally, creating a hydrogeological numerical model requires defining the temporal and spatial properties of the model, including setting the starting time and the vertical, horizontal, and lateral boundaries. The continuous space is divided into finite blocks, and the continuous time is divided into finite time steps. The physical and mechanical properties of the geological formations are then specified, followed by designating the groundwater seepage conditions as a function of time. After running the model, a new water level distribution is calculated for each time step based on the combi nation of water level, prerequisite, and boundary conditions. In constructing numerical models for multi-year permafrost regions, the seasonal changes in active layer geological stratum attributes, the insulating effects of snow cover on the soil surface, and soil moisture changes resulting from snow-melt infiltration make the modeling process more complex. Frampton employed a two-dimensional water–heat coupling model to study the processes of groundwater seepage and discharge in multi-year permafrost regions. The modeling results suggest that as the thickness of the island-shaped active layer in these regions increases, more groundwater infiltrates into deeper aquifers and is later expelled onto the surface, thereby amplifying the depth and length of the groundwater flow paths. Breemer used the MODFLOW groundwater model to simulate the two-dimensional groundwater seepage processes in the Lake Michigan Lobe region, and achieved simulation results for steady-state hydraulic head values and surface drainage volumes that were comparable to those measured in the field. Wellman utilized simulations to analyze the two-dimensional groundwater seepage processes in the talik zone of Alaska's interior Yukon region, and demonstrated how time, climate, lake size, and hydraulic conditions influenced the development of taliks and variations in water content within the aquifer. (in red)
- Riseborough, D.; Shiklomanov, N.; Etzelmüller, B.; Gruber, S.; Marchenko, S. Recent advances in permafrost modelling. Permafr. Periglac. Process 2008, 19, 137-156, doi:10.1002/ppp.615
- Woo, M.K.; Kane, D.L.; Carey, S.K.; Yang, D. Progress in permafrost hydrology in the new millennium. Permafr. Periglac. Process 2008, 19, 237-254, doi:10.1002/ppp.613.
- Frampton, A.; Painter, S.L.; Destouni, G. Permafrost degradation and subsurface-flow changes caused by surface warming trends. Hydrogeol. J. 2013, 21, 271, doi:10.1007/s10040-012-0938-z
- Koch, J.C.; McKnight, D.M.; Neupauer, R.M. Simulating unsteady flow, anabranching, and hyporheic dynamics in a glacial meltwater stream using a coupled surface water routing and groundwater flow model. Water Resour. Res. 2011, 47, doi:10.1029/2010wr009508.
- Ge, S.; McKenzie, J.; Voss, C.; Wu, Q. Exchange of groundwater and surface-water mediated by permafrost response to seasonal and long term air temperature variation. Geophys. Res. Lett. 2011, 38, L14402, doi:10.1029/2011gl047911.
- Gaiolini, M.; Colombani, N.; Busico, G.; Rama, F.; Mastrocicco, M. Impact of Boundary Conditions Dynamics on Groundwater Budget in the Campania Region (Italy). Water 2022, 14, 2462, doi:10.3390/w14162462
- Woo, M.-k.; Mollinga, M.; Smith, S.L. Modeling maximum active layer thaw in boreal and tundra environments using limited data; Springer-Verlag Berlin Heidelberg: Heidelberg, 2008; pp. 125-137.
- Oelke, C.; Zhang, T. A model study of circum-Arctic soil temperatures. Permafr. Periglac. Process 2004, 15, 103-121, doi:10.1002/ppp.485.
- Zhang, T. Influence of the seasonal snow cover on the ground thermal regime: An overview. Rev. Geophys. 2005, 43, RG4002, doi:10.1029/2004rg000157.
- Iwata, Y.; Hayashi, M.; Hirota, T. Comparison of snowmelt infiltration under different soil-freezing conditions influenced by snow cover. Vadose Zone J. 2008, 7, 79-86, doi:10.2136/vzj2007.0089.
- Frampton, A.; Painter, S.; Lyon, S.W.; Destouni, G. Non-isothermal, three-phase simulations of near-surface flows in a model permafrost system under seasonal variability and climate change. J. Hydrol. 2011, 403, 352-359, doi:10.1016/j.jhydrol.2011.04.010.
- Breemer, C. Glacier Science and Environmental Change; Blackwell Publishing: Hoboken, USA, 2006; pp. 63-66.
- Wellman, T.P.; Voss, C.I.; Walvoord, M.A. Impacts of climate, lake size, and supra-and sub-permafrost groundwater flow on lake-talik evolution, Yukon Flats, Alaska (USA). Hydrogeol. J. 2013, 21, 281, doi:10.1007/s10040-012-0941-4.

Reviewer 2 Report
The topic of the article is interesting and stimulating. Here you can find my comments:
OVERALL: - Please, enlarge and re-arrange font sizes to guide the reader properly in all sections. All figures must be composed of HD images. It is mandatory to improve the scientific quality of the whole manuscript.
- Please, pay attention to the JOURNAL TEMPLATE within the entire manuscript: in all sections, including tables, references, captions, units, equations, and Figures.
INTRODUCTION: Please, consider in the scientific background of your study the value of both advanced experimental and modelling analysis of the prediction of phenomena.
I SUGGEST MINOR REVISIONS
Here the comments:
Line 50-54: Insert references.
Lines 55-56: Insert reference
Line 60: “permafrost”
Line 78: Add some examples about the use of MODFLOW in other type of study. For example: DOI: 10.3390/w14162462.
Lines 125-129: Add the unite of measurement for the equation terms
Figure 2: Improve the quality of the figure. Maybe is better to put the legend outside the map.
Figure 4: improve the quality of the figure increasing the size of the text.
Line 189: Add some references for the used equations.
Figure 5: Improve the figure quality. The axis names of the small graph are not readable. Put the legend outside the graph.
Figure 6: The quality is dramatically low. Increase the labels text and the legend text.
Figure 7: Increase the quality of the figure. Increase the text because is not readable. Please add a,b,c etc. to each map of the figure.
Figure 8: Same problems of figure 7. I you write in the text “part b and c, figure 8” you must indicate in the figures which are the parts a,b and c.
I highly recommend that you carefully review the English form of your paper and make necessary corrections to any writing errors. As a language model, I understand the importance of proper grammar, spelling, and punctuation in written communication. These mistakes can not only detract from the overall quality of your paper, but they can also create confusion for readers and potentially impact the credibility of your work. To ensure that your paper is presented in the best possible way, I suggest that you take the time to carefully review and edit your writing.
Author Response
Response to Reviewer 2 Comments
We feel great thanks for your professional review work on our article. As you are concerned, there are several problems that need to be addressed. According to your nice suggestions, we have made extensive corrections to our previous draft, the detailed corrections are listed below.
Point 1: The topic of the article is interesting and stimulating. Here you can find my comments:
OVERALL: - Please, enlarge and re-arrange font sizes to guide the reader properly in all sections. All figures must be composed of HD images. It is mandatory to improve the scientific quality of the whole manuscript.
Please, pay attention to the JOURNAL TEMPLATE within the entire manuscript: in all sections, including tables, references, captions, units, equations, and Figures.
INTRODUCTION: Please, consider in the scientific background of your study the value of both advanced experimental and modelling analysis of the prediction of phenomena.
I SUGGEST MINOR REVISIONS
Response 1: We feel great thanks for your professional review work on our article. According to your nice suggestions, we have made extensive corrections to our previous draft, the detailed corrections are listed below. All Figures throughout the paper have been updated to improve their quality. We also expanded our discussion of numerical simulation in the introduction and placed our study in context with previously published research. In addition to this, we meticulously checked and revised all details of the manuscript to comply with journal guidelines. (in red)
Point 2: Line 50-54: Insert references.
Response 2: Thank you for pointing this out. Line 59-63: As suggested by the reviewer, we have added reference to support this idea. (in red)
Jin, H.; Huang, Y.; Bense, V.F.; Ma, Q.; Marchenko, S.S.; Shepelev, V.V.; Hu, Y.; Liang, S.; Spektor, V.V.; Jin, X.; et al. Permafrost degradation and its hydrogeological impacts. Water 2022, 14, 372, doi:10.3390/w14030372.
Point 3: Lines 55-56: Insert reference
Response 3: Thank you for pointing this out. Line 64-66: As suggested by the reviewer, we have added reference to support this idea. (in red)
Li, J.; Chen, W.; Liu, Z.; Li, J.; Chen, W.; Liu, Z. The Qinghai–Tibet Railway Geological Environment; Springer Berlin: Heidelberg, German, 2018; pp. 11-71.
Point 4: Line 60: “permafrost”
Response 4: Line 69: We sincerely thank the reviewer for careful reading. As suggested by the reviewer, we have corrected the “p ermafrost” into “permafrost”. (in red)
Point 5: Line 78: Add some examples about the use of MODFLOW in other type of study. For example: DOI: 10.3390/w14162462.
Response 5: Line 78-105: Within the introduction section, we included more detailed discussion surrounding numerical simulation and positioned our study in relation to previously published literature. Additionally, we bolstered our work with 13 supplementary references. The revised text reads as follows.
The hydrogeological numerical model is highly adaptable, and can accommodate varying geological attributes, geometries, and boundary conditions. Therefore, the use of numerical simulations to assess the characteristics of aquifers in permafrost zones has received increasing attention.
Generally, creating a hydrogeological numerical model requires defining the temporal and spatial properties of the model, including setting the starting time and the vertical, horizontal, and lateral boundaries. The continuous space is divided into finite blocks, and the continuous time is divided into finite time steps. The physical and mechanical properties of the geological formations are then specified, followed by designating the groundwater seepage conditions as a function of time. After running the model, a new water level distribution is calculated for each time step based on the combi nation of water level, prerequisite, and boundary conditions. In constructing numerical models for multi-year permafrost regions, the seasonal changes in active layer geological stratum attributes, the insulating effects of snow cover on the soil surface, and soil moisture changes resulting from snow-melt infiltration make the modeling process more complex. Frampton employed a two-dimensional water–heat coupling model to study the processes of groundwater seepage and discharge in multi-year permafrost regions. The modeling results suggest that as the thickness of the island-shaped active layer in these regions increases, more groundwater infiltrates into deeper aquifers and is later expelled onto the surface, thereby amplifying the depth and length of the groundwater flow paths. Breemer used the MODFLOW groundwater model to simulate the two-dimensional groundwater seepage processes in the Lake Michigan Lobe region, and achieved simulation results for steady-state hydraulic head values and surface drainage volumes that were comparable to those measured in the field. Wellman utilized simulations to analyze the two-dimensional groundwater seepage processes in the talik zone of Alaska's interior Yukon region, and demonstrated how time, climate, lake size, and hydraulic conditions influenced the development of taliks and variations in water content within the aquifer. (in red)
- Gaiolini, M.; Colombani, N.; Busico, G.; Rama, F.; Mastrocicco, M. Impact of Boundary Conditions Dynamics on Groundwater Budget in the Campania Region (Italy). Water 2022, 14, 2462, doi:10.3390/w14162462
- Riseborough, D.; Shiklomanov, N.; Etzelmüller, B.; Gruber, S.; Marchenko, S. Recent advances in permafrost modelling. Permafr. Periglac. Process 2008, 19, 137-156, doi:10.1002/ppp.615
- Woo, M.K.; Kane, D.L.; Carey, S.K.; Yang, D. Progress in permafrost hydrology in the new millennium. Permafr. Periglac. Process 2008, 19, 237-254, doi:10.1002/ppp.613.
- Frampton, A.; Painter, S.L.; Destouni, G. Permafrost degradation and subsurface-flow changes caused by surface warming trends. Hydrogeol. J. 2013, 21, 271, doi:10.1007/s10040-012-0938-z
- Koch, J.C.; McKnight, D.M.; Neupauer, R.M. Simulating unsteady flow, anabranching, and hyporheic dynamics in a glacial meltwater stream using a coupled surface water routing and groundwater flow model. Water Resour. Res. 2011, 47, doi:10.1029/2010wr009508.
- Ge, S.; McKenzie, J.; Voss, C.; Wu, Q. Exchange of groundwater and surface-water mediated by permafrost response to seasonal and long term air temperature variation. Geophys. Res. Lett. 2011, 38, L14402, doi:10.1029/2011gl047911.
- Woo, M.-k.; Mollinga, M.; Smith, S.L. Modeling maximum active layer thaw in boreal and tundra environments using limited data; Springer-Verlag Berlin Heidelberg: Heidelberg, 2008; pp. 125-137.
- Oelke, C.; Zhang, T. A model study of circum-Arctic soil temperatures. Permafr. Periglac. Process 2004, 15, 103-121, doi:10.1002/ppp.485.
- Zhang, T. Influence of the seasonal snow cover on the ground thermal regime: An overview. Rev. Geophys. 2005, 43, RG4002, doi:10.1029/2004rg000157.
- Iwata, Y.; Hayashi, M.; Hirota, T. Comparison of snowmelt infiltration under different soil-freezing conditions influenced by snow cover. Vadose Zone J. 2008, 7, 79-86, doi:10.2136/vzj2007.0089.
- Frampton, A.; Painter, S.; Lyon, S.W.; Destouni, G. Non-isothermal, three-phase simulations of near-surface flows in a model permafrost system under seasonal variability and climate change. J. Hydrol. 2011, 403, 352-359, doi:10.1016/j.jhydrol.2011.04.010.
- Breemer, C. Glacier Science and Environmental Change; Blackwell Publishing: Hoboken, USA, 2006; pp. 63-66.
- Wellman, T.P.; Voss, C.I.; Walvoord, M.A. Impacts of climate, lake size, and supra-and sub-permafrost groundwater flow on lake-talik evolution, Yukon Flats, Alaska (USA). Hydrogeol. J. 2013, 21, 281, doi:10.1007/s10040-012-0941-4.
Point 6: Lines 125-129: Add the unite of measurement for the equation terms.
Response 6: We sincerely thank the reviewer for careful reading. Lines 177-181: We add the unite of measurement for the equation terms. X is the temperature or precipitation of the study area (℃ or mm); Di is the horizontal distance between the interpolated hydrometeorological station and the study area (m); Xi is the temperature or precipitation of the interpolated hydrometeorological station (℃ or mm); n is the number of interpolated sample points; and p is the power exponent used to calculate the weight of dis tance. (in red)
Point 7: Figure 2: Improve the quality of the figure. Maybe is better to put the legend outside the map.
Response 7: Figure 2: We think this is an excellent suggestion. We have improved the quality of the figure. We place the legend outside the map. Please refer see the attachment. (in red)
Point 8: Figure 4: improve the quality of the figure increasing the size of the text.
Response 8: We think this is an excellent suggestion. Figure 4: We have improved the quality of the figure and increased the size of the text. Please refer see the attachment. (in red)
Point 9: Line 189: Add some references for the used equations.
Response 9: We were really sorry for our careless mistakes. Thank you for your reminder. Line 262: We add references for the used equations. (in red)
Harbaugh, A.W. MODFLOW-2005, the US Geological Survey modular ground-water model: the ground-water flow process; US Department of the Interior, US Geological Survey Reston, VA, USA: 2005; Volume 6.
Point 10: Figure 5: Improve the figure quality. The axis names of the small graph are not readable. Put the legend outside the graph.
Response 10: We have improved the quality of the figure, enlarged the names of the axes in the smaller graph and placed the legends outside the figure. Thank you very much for your suggestions, which have greatly improved the readability of the figure. Please refer see the attachment. (in red)
Point 11: Figure 6: The quality is dramatically low. Increase the labels text and the legend text.
Response 11: Thank you for pointing this out. Figure 3 and Figure 4: We have recreated the geological cross-section and incorporated two additional cross-sections from the resistivity tomography scanning, ultimately leading to a more defined range of the taliks. Please refer see the attachment. (in red)
Point 12: Figure 7: Increase the quality of the figure. Increase the text because is not readable. Please add a,b,c etc. to each map of the figure.
Response 12: We think this is an excellent suggestion. Figure 8: We have improved the quality of each image and added a number (a, b, c etc) to each one. Please refer see the attachment. (in red)
Point 13: Figure 8: Same problems of figure 7. I you write in the text “part b and c, figure 8” you must indicate in the figures which are the parts a,b and c.
Response 13: Once again, we apologise for the oversight. Figure 9: We have improved the quality of each image and added a number (a, b, c etc) to each one. Please refer see the attachment. (in red)
Point 14: I highly recommend that you carefully review the English form of your paper and make necessary corrections to any writing errors. As a language model, I understand the importance of proper grammar, spelling, and punctuation in written communication. These mistakes can not only detract from the overall quality of your paper, but they can also create confusion for readers and potentially impact the credibility of your work. To ensure that your paper is presented in the best possible way, I suggest that you take the time to carefully review and edit your writing.
Response 14: Thanks for your suggestion. We feel sorry for our poor writings, however, we do invite a friend of us who is a native English speaker from the USA to help polish our article. And we hope the revised manuscript could be acceptable for you. (in red)

Reviewer 3 Report
The authors used the MODFLOW-USG model, a three-dimensional standard groundwater seepage model for quantitatively assessing the dynamic characteristics of groundwater in the Eruu area in Central Yakutia. The manuscript needs extensive revisions as it is written very poorly and very difficult to understand for readers.
Major Comments:
The literature section is weak and authors should use extensive literature to relate the current studiey with already published studies.
There should be a discussion section highlighting findings of the current study with previous studies published in a similar area or different areas.
Minor Comments:
Line 19-24: This sentence is very long. Please rephrase it.
Line 24-25: Please clarify what “well 14E/2014” is showing.
Line 113: It should be mentioned the spatial resolution of DEM whether it is satellite-based or some other source.
The English language used needs editing throughout with a moderate level of change required as there are many grammatical errors.
Author Response
Response to Reviewer 3 Comments
According to the reviewer’s comments, we have revised the manuscript extensively. If there are any other modifications we could make, we would like very much to modify them and we really appreciate your help.
Point 1: The literature section is weak and authors should use extensive literature to relate the current studiey with already published studies.
There should be a discussion section highlighting findings of the current study with previous studies published in a similar area or different areas.
Response 1: I would like to express our sincere gratitude to the reviewers for their comments. Your comment arc all valuable and helpful for revising and improving our manuscript, as well as the important guiding significance to our researches.
Line 78-105: Within the introduction section, we included more detailed discussion surrounding numerical simulation and positioned our study in relation to previously published literature. Additionally, we bolstered our work with 13 supplementary references. The revised text reads as follows.
The hydrogeological numerical model is highly adaptable, and can accommodate varying geological attributes, geometries, and boundary conditions. Therefore, the use of numerical simulations to assess the characteristics of aquifers in permafrost zones has received increasing attention.
Generally, creating a hydrogeological numerical model requires defining the temporal and spatial properties of the model, including setting the starting time and the vertical, horizontal, and lateral boundaries. The continuous space is divided into finite blocks, and the continuous time is divided into finite time steps. The physical and mechanical properties of the geological formations are then specified, followed by designating the groundwater seepage conditions as a function of time. After running the model, a new water level distribution is calculated for each time step based on the combi nation of water level, prerequisite, and boundary conditions. In constructing numerical models for multi-year permafrost regions, the seasonal changes in active layer geological stratum attributes, the insulating effects of snow cover on the soil surface, and soil moisture changes resulting from snow-melt infiltration make the modeling process more complex. Frampton employed a two-dimensional water–heat coupling model to study the processes of groundwater seepage and discharge in multi-year permafrost regions. The modeling results suggest that as the thickness of the island-shaped active layer in these regions increases, more groundwater infiltrates into deeper aquifers and is later expelled onto the surface, thereby amplifying the depth and length of the groundwater flow paths. Breemer used the MODFLOW groundwater model to simulate the two-dimensional groundwater seepage processes in the Lake Michigan Lobe region, and achieved simulation results for steady-state hydraulic head values and surface drainage volumes that were comparable to those measured in the field. Wellman utilized simulations to analyze the two-dimensional groundwater seepage processes in the talik zone of Alaska's interior Yukon region, and demonstrated how time, climate, lake size, and hydraulic conditions influenced the development of taliks and variations in water content within the aquifer. (in red)
- Riseborough, D.; Shiklomanov, N.; Etzelmüller, B.; Gruber, S.; Marchenko, S. Recent advances in permafrost modelling. Permafr. Periglac. Process 2008, 19, 137-156, doi:10.1002/ppp.615
- Woo, M.K.; Kane, D.L.; Carey, S.K.; Yang, D. Progress in permafrost hydrology in the new millennium. Permafr. Periglac. Process 2008, 19, 237-254, doi:10.1002/ppp.613.
- Frampton, A.; Painter, S.L.; Destouni, G. Permafrost degradation and subsurface-flow changes caused by surface warming trends. Hydrogeol. J. 2013, 21, 271, doi:10.1007/s10040-012-0938-z
- Koch, J.C.; McKnight, D.M.; Neupauer, R.M. Simulating unsteady flow, anabranching, and hyporheic dynamics in a glacial meltwater stream using a coupled surface water routing and groundwater flow model. Water Resour. Res. 2011, 47, doi:10.1029/2010wr009508.
- Ge, S.; McKenzie, J.; Voss, C.; Wu, Q. Exchange of groundwater and surface-water mediated by permafrost response to seasonal and long term air temperature variation. Geophys. Res. Lett. 2011, 38, L14402, doi:10.1029/2011gl047911.
- Gaiolini, M.; Colombani, N.; Busico, G.; Rama, F.; Mastrocicco, M. Impact of Boundary Conditions Dynamics on Groundwater Budget in the Campania Region (Italy). Water 2022, 14, 2462, doi:10.3390/w14162462
- Woo, M.-k.; Mollinga, M.; Smith, S.L. Modeling maximum active layer thaw in boreal and tundra environments using limited data; Springer-Verlag Berlin Heidelberg: Heidelberg, 2008; pp. 125-137.
- Oelke, C.; Zhang, T. A model study of circum-Arctic soil temperatures. Permafr. Periglac. Process 2004, 15, 103-121, doi:10.1002/ppp.485.
- Zhang, T. Influence of the seasonal snow cover on the ground thermal regime: An overview. Rev. Geophys. 2005, 43, RG4002, doi:10.1029/2004rg000157.
- Iwata, Y.; Hayashi, M.; Hirota, T. Comparison of snowmelt infiltration under different soil-freezing conditions influenced by snow cover. Vadose Zone J. 2008, 7, 79-86, doi:10.2136/vzj2007.0089.
- Frampton, A.; Painter, S.; Lyon, S.W.; Destouni, G. Non-isothermal, three-phase simulations of near-surface flows in a model permafrost system under seasonal variability and climate change. J. Hydrol. 2011, 403, 352-359, doi:10.1016/j.jhydrol.2011.04.010.
- Breemer, C. Glacier Science and Environmental Change; Blackwell Publishing: Hoboken, USA, 2006; pp. 63-66.
- Wellman, T.P.; Voss, C.I.; Walvoord, M.A. Impacts of climate, lake size, and supra-and sub-permafrost groundwater flow on lake-talik evolution, Yukon Flats, Alaska (USA). Hydrogeol. J. 2013, 21, 281, doi:10.1007/s10040-012-0941-4.
We have added a discussion section. The revised text reads as follows.
Line: 462-481: This study utilized the MODFLOW-USG model to evaluate the dynamic changes in groundwater levels in Eruu, Sakha (Yakutia) Republic, Russia. Considering that the research area is situated in a region where there is permafrost, the seasonal freeze-thaw processes within the active layer and the infiltration of snowmelt water into the soil during late spring may have an impact on groundwater levels. To address these impacts comprehensively, the present study undertook a thorough examination of these factors. The issue of dynamic changes in groundwater in permafrost regions has been receiving increasing attention, with some scholars conducting 2D simulation studies to address this problem. With the installation of 11 hydrogeological monitoring wells in the study area, we gathered abundant geological data. Building on a profound understanding of boundary conditions, we established a 3D groundwater flow model that is vital to the implementation of sustainable water resource management. Nonetheless, incomplete data caused by the harsh temperature conditions and remote location of the study area were an issue. Thus, we aim to supplement our data further using other water level monitoring methods, including long-term manual observations. (in red)
Point 2: Line 19-24: This sentence is very long. Please rephrase it.
Response 2: Line 19-24: Thank you for pointing this out. We have revised this very long sentence to improve the readability of the phrase. Detailed corrections are listed below.
The perennially unfrozen zones (taliks) in the Eruu area of central Yakutia have a complex stratigraphic structure, and the dynamic characteristics of groundwater in this region have been insufficiently studied. This study analyzed the results of the explorations and geophysical studies conducted by the Melnikov Permafrost Institute of the Siberian Branch of the Russian Academy of Science. In addition, we simulated and analyzed the dynamic characteristics of groundwater in the area based on hydro-meteorological data, snow data, and remote sensing data. During the process, the dynamic changes in the attributes of aquifers due to the seasonal freeze–thaw processes of soils, including the active layer, were also taken into account. (in red)
Point 3: Line 24-25: Please clarify what “well 14E/2014” is showing.
Response 3: Line 32-33: Thank you for pointing this out. As presented in Figure 2, 14E/2014 refers to a hydrogeological observation well. To assess the accuracy of simulated groundwater levels from September 1, 2014 to August 31, 2019, we selected well 14E/2014 due to its more complete and accurate measured data. Our results reveal that more than 99% of the data points have an error value of less than 0.1m, which is illustrated in red in Figure 7. Please refer see the attachment.
Point 4: Line 113: It should be mentioned the spatial resolution of DEM whether it is satellite-based or some other source.
Response 4: Line 158-162: The DEM data used in this study were obtained from the ASTER GDEM V3 dataset published by NASA in 2019. The data have a spatial resolution of approximately 30m, and were calibrated using actual topographic data measured by the Melnikov Permafrost Institute of the Siberian Branch of the Russian Academy of Science. (in red)

Reviewer 4 Report
The manuscript titled as " Simulation and Analysis of the Dynamic Characteristics of Groundwater in Taliks in the Eruu Area, Central Yakutia" is aimed to study the dynamic characteristic of groundwater in the perennially unfrozen zones. The topic is interesting and valuable for the Reader of the Sustainability. However, I feel that the paper needs to be minorly revised before it can be accepted. I have reviewed this paper seriously and presented the main comments below.
1. The authors should introduce more information of the aquifers, such as thickness, lithology and so on.
2. Provide geological section of the study area.
3. Provide information of groundwater exploitation.
4. There is Only one monitoring well that is available for comparing simulated and observed water levels in this paper. It not enough. Provide more comparison of simulated and observed water levels.
5. Provide a comparison of simulated and observed groundwater flow field.
no.
Author Response
Response to Reviewer 4 Comments
Thank you for your positive comments and valuable suggestions to improve the quality of our manuscript. If there are any other modifications we could make, we would like very much to modify them and we really appreciate your help. Thank you very much for your help.
Point 1: The authors should introduce more information of the aquifers, such as thickness, lithology and so on.
Response 1: Thank you for pointing this out. The reviewer is correct, and we have added more aquifer information. The revised text reads as follows.
Line138-149: In 2007, the hydrogeological observation well 1-2007 was drilled within the groundwater recharge zone between Ergen Lake and Abaga-Quill Lake. Interlayer water in the frozen layer was found to be pressurized at depths of 23.9-32.0m, and water samples collected from the lakes, talik aquifer, and Eruu spring were found to be chemically identical. In 2014, the hydrogeological observation well 14E was drilled between Abaga-Quill Lake and Eruu spring. Beneath the permanent permafrost at a depth of 13m, water-saturated sand at 0.2℃ was discovered. At the point where the groundwater was discharged, the thickness of the aquifer was determined to be 23-30m. The aquifer consisted primarily of fine and medium-grained sand of the Quaternary period, while the aquifer at the springwater edge was composed of Quaternary sandstone, gravel and pebble deposits, and upper Jurassic sandstone. (in red)
Point 2: Provide geological section of the study area.
Response 2: We feel great thanks for your point out. We have redrawn the hydrogeological section of the study area (Figure 4) and supplemented two geophysical observation profiles in the study area (Figure 3). We determine the presence and approximate size of taliks.
Line219-228: We conducted electrical resistivity tomography scanning to survey the profiles of two taliks in the study area, and the results are presented in Figure 3. Within the surveyed sections, the measured electrical resistivity ranged from 50 to 50,000Ω·m. Combined with previous experimental data, the unfrozen soil layers had an electrical resistivity range of 50-2000Ω·m. The electrical resistivity range of frozen soil layers at a temperature near -0.2℃ was 2000-10000Ω·m, while the electrical resistivity range of permafrost at lower temperatures was 10000-50000Ω·m. The water-bearing layer detected in profile A1-A2 had an estimated width of about 650m, with a top depth of 13-30m. The water-bearing layer detected in profile B1-B2 had an estimated width of about 500m, with a top depth of 20-34m. Please refer see the attachment. (in red)
Point 3: Provide information of groundwater exploitation.
Response 3: Thank you for pointing this out. Despite favorable groundwater quality in the research area, it has not been extracted due to its distance from residential areas and the difficulty of exploitation. Yakut considers this area mainly as a reserve water source. (in red)
Point 4: There is Only one monitoring well that is available for comparing simulated and observed water levels in this paper. It not enough. Provide more comparison of simulated and observed water levels.
Response 4: We agree that more comparisons between simulated and observed water levels would significantly improve the quality of the article. We drilled 11 hydrogeological observation wells in the study area. However, only one of these wells, named 14E/2014 and drilled in 2014, operated efficiently throughout the research period. The persistent low-temperature environment in the region posed significant challenges to regular automatic monitoring of water levels. As a result, the other wells were either damaged or had a shorter operating time, occurring at various periods in other years. Furthermore, since the research area is quite far from the residential region and the route is rugged, it is challenging to use large equipment for the study. Hence, we intend to explore other water level observation approaches, such as long-term manual observations, to supplement our data. (in red)
Point 5: Provide a comparison of simulated and observed groundwater flow field.
Response 5: Thank you again for your positive comments and valuable suggestions to improve the quality of our manuscript. Unfortunately, our attempts to obtain more data about the research area have been hindered by the presence of permafrost, making it challenging to gather sufficient information. Nonetheless, despite the data deficiency, the high quality of the groundwater in the area, which can be directly consumed, makes it a valuable resource. This is one of the driving factors behind the development of our modeling methodology. To make up for the lack of data, we are taking steps to include additional monitoring methods for the groundwater seepage and water level data. (in red)

Reviewer 5 Report
In the manuscript titled Simulation and Analysis of the Dynamic Characteristics of Groundwater in Taliks in the Eruu Area, Central Yakutia(Manuscript ID: sustainability-2352201), the authors simulated and analyzed the dynamic characteristics of groundwater in Taliks in the permafrost region. The model took into account the dynamic changes in aquifer attributes due to seasonal freeze-thaw processes in the soil. It has good simulation results and the error is within 0.1m. This study provides an effective three-dimensional modeling method for quantifying the analysis of groundwater dynamics in permafrost regions. However, there are still some parts that need further improvement. Therefore, minor revision has to be completed before this manuscript could be accepted for publication.
Comment 1: The authors should add some of the latest studies pertinent to theoretical support in the introduction.
Comment 2: Line 246, the source of the data for the net radiation index Rd needs to be added. The formula is incorrectly numbered.
Comment 3: Line 311, The authors believe that the water level in well 14E/2014 was strongly influenced by Lake Abaga Quel and suggests further explanation.
Comment 4: Table 1 and Table 2 respectively show the parameter ranges of different geological strata. It is recommended to retain the optimized parameter ranges.
Comment 5: Line 353 and the third article in the conclusion mentioned that "The mean value of simulated discharge during the period was at 3888.39 L/d", which should only be the excretion of spring, please explain it accurately.
Comment 6: Proofread of the paper is required.
Author Response
Response to Reviewer 5 Comments
I would like to express our sincere gratitude to the reviewers for their comments. Your comments arc all valuable and helpful for revising and improving our manuscript, as well as the important guiding significance to our researches.
Point 1: The authors should add some of the latest studies pertinent to theoretical support in the introduction.
Response 1: We feel great thanks for your professional review work on our article. Line 78-105: Within the introduction section, we included more detailed discussion surrounding numerical simulation and positioned our study in relation to previously published literature. Additionally, we bolstered our work with 13 supplementary references. The revised text reads as follows.
The hydrogeological numerical model is highly adaptable, and can accommodate varying geological attributes, geometries, and boundary conditions. Therefore, the use of numerical simulations to assess the characteristics of aquifers in permafrost zones has received increasing attention.
Generally, creating a hydrogeological numerical model requires defining the temporal and spatial properties of the model, including setting the starting time and the vertical, horizontal, and lateral boundaries. The continuous space is divided into finite blocks, and the continuous time is divided into finite time steps. The physical and mechanical properties of the geological formations are then specified, followed by designating the groundwater seepage conditions as a function of time. After running the model, a new water level distribution is calculated for each time step based on the combi nation of water level, prerequisite, and boundary conditions. In constructing numerical models for multi-year permafrost regions, the seasonal changes in active layer geological stratum attributes, the insulating effects of snow cover on the soil surface, and soil moisture changes resulting from snow-melt infiltration make the modeling process more complex. Frampton employed a two-dimensional water–heat coupling model to study the processes of groundwater seepage and discharge in multi-year permafrost regions. The modeling results suggest that as the thickness of the island-shaped active layer in these regions increases, more groundwater infiltrates into deeper aquifers and is later expelled onto the surface, thereby amplifying the depth and length of the groundwater flow paths. Breemer used the MODFLOW groundwater model to simulate the two-dimensional groundwater seepage processes in the Lake Michigan Lobe region, and achieved simulation results for steady-state hydraulic head values and surface drainage volumes that were comparable to those measured in the field. Wellman utilized simulations to analyze the two-dimensional groundwater seepage processes in the talik zone of Alaska's interior Yukon region, and demonstrated how time, climate, lake size, and hydraulic conditions influenced the development of taliks and variations in water content within the aquifer. (in red)
- Riseborough, D.; Shiklomanov, N.; Etzelmüller, B.; Gruber, S.; Marchenko, S. Recent advances in permafrost modelling. Permafr. Periglac. Process 2008, 19, 137-156, doi:10.1002/ppp.615
- Woo, M.K.; Kane, D.L.; Carey, S.K.; Yang, D. Progress in permafrost hydrology in the new millennium. Permafr. Periglac. Process 2008, 19, 237-254, doi:10.1002/ppp.613.
- Frampton, A.; Painter, S.L.; Destouni, G. Permafrost degradation and subsurface-flow changes caused by surface warming trends. Hydrogeol. J. 2013, 21, 271, doi:10.1007/s10040-012-0938-z
- Koch, J.C.; McKnight, D.M.; Neupauer, R.M. Simulating unsteady flow, anabranching, and hyporheic dynamics in a glacial meltwater stream using a coupled surface water routing and groundwater flow model. Water Resour. Res. 2011, 47, doi:10.1029/2010wr009508.
- Ge, S.; McKenzie, J.; Voss, C.; Wu, Q. Exchange of groundwater and surface-water mediated by permafrost response to seasonal and long term air temperature variation. Geophys. Res. Lett. 2011, 38, L14402, doi:10.1029/2011gl047911.
- Gaiolini, M.; Colombani, N.; Busico, G.; Rama, F.; Mastrocicco, M. Impact of Boundary Conditions Dynamics on Groundwater Budget in the Campania Region (Italy). Water 2022, 14, 2462, doi:10.3390/w14162462
- Woo, M.-k.; Mollinga, M.; Smith, S.L. Modeling maximum active layer thaw in boreal and tundra environments using limited data; Springer-Verlag Berlin Heidelberg: Heidelberg, 2008; pp. 125-137.
- Oelke, C.; Zhang, T. A model study of circum-Arctic soil temperatures. Permafr. Periglac. Process 2004, 15, 103-121, doi:10.1002/ppp.485.
- Zhang, T. Influence of the seasonal snow cover on the ground thermal regime: An overview. Rev. Geophys. 2005, 43, RG4002, doi:10.1029/2004rg000157.
- Iwata, Y.; Hayashi, M.; Hirota, T. Comparison of snowmelt infiltration under different soil-freezing conditions influenced by snow cover. Vadose Zone J. 2008, 7, 79-86, doi:10.2136/vzj2007.0089.
- Frampton, A.; Painter, S.; Lyon, S.W.; Destouni, G. Non-isothermal, three-phase simulations of near-surface flows in a model permafrost system under seasonal variability and climate change. J. Hydrol. 2011, 403, 352-359, doi:10.1016/j.jhydrol.2011.04.010.
- Breemer, C. Glacier Science and Environmental Change; Blackwell Publishing: Hoboken, USA, 2006; pp. 63-66.
- Wellman, T.P.; Voss, C.I.; Walvoord, M.A. Impacts of climate, lake size, and supra-and sub-permafrost groundwater flow on lake-talik evolution, Yukon Flats, Alaska (USA). Hydrogeol. J. 2013, 21, 281, doi:10.1007/s10040-012-0941-4.
Point 2: Line 246, the source of the data for the net radiation index Rd needs to be added. The formula is incorrectly numbered.
Response 2: We sincerely thank the reviewer for careful reading. Line 162-163: Solar radiation data were obtained from the Pokrovsk hydrometeorological station, which is the closest to the study area. We have corrected all formula numbers. (in red)
Point 3: Line 311, The authors believe that the water level in well 14E/2014 was strongly influenced by Lake Abaga Quel and suggests further explanation.
Response 3: Thank you for pointing this out. We would like to explain this further to you.Because there are two main reasons for the change in water level affecting Well 14E/2014. One is the recharge of groundwater in talik, and the other is the recharge of groundwater in Lake Abaga-Quill. From 2007 to 2012, we observed the water level of Well 1/2007. The results show that the water level in talik changes very little in the same period of the year, so the analysis points out that the groundwater recharge of Abaga-Quill Lake is the main reason affecting the water level change of No. 14E/2014 well. (in red)
Point 4: Table 1 and Table 2 respectively show the parameter ranges of different geological strata. It is recommended to retain the optimized parameter ranges.
Response 4: Thanks for your suggestion. Line 363: We have retained the optimized parameter ranges. (in red)
Point 5: Line 353 and the third article in the conclusion mentioned that "The mean value of simulated discharge during the period was at 3888.39 L/d", which should only be the excretion of spring, please explain it accurately.
Response 5: We sincerely thank the reviewer for careful reading. Detailed corrections are listed below. Line 41 and Line 498: “The mean value of simulated discharge from the spring during the period was at 3888.39 L/d.” (in red)
Point 6: Proofread of the paper is required.
Response 6: We feel great thanks for your professional review work on our article. According to your nice suggestions, we have made extensive corrections to our previous draft, the detailed corrections are listed below. All figures throughout the paper have been updated to improve their quality. We also expanded our discussion of numerical simulation in the introduction and placed our study in context with previously published research. In addition to this, we meticulously checked and revised all details of the manuscript to comply with journal guidelines. (in red)

Round 2
Reviewer 3 Report
The manuscript has been improved substanitally and it should be published.